# Metabolic Stress and Mitochondrial Dysfunction in Ataxia-Telangiectasia

**DOI:** 10.3390/antiox11040653

**Published:** 2022-03-28

**Authors:** Goutham Narayanan Subramanian, Abrey Jie Yeo, Magtouf Hnaidi Gatei, David John Coman, Martin Francis Lavin

**Affiliations:** 1University of Queensland Centre for Clinical Research, University of Queensland, Brisbane, QLD 4029, Australia; g.subramanian@uq.edu.au (G.N.S.); abrey.yeo@uq.edu.au (A.J.Y.); m.gatei@uq.edu.au (M.H.G.); 2Queensland Children’s Hospital, Brisbane, QLD 4101, Australia; d.coman@uq.edu.au; 3Faculty of Medicine, University of Queensland, Brisbane, QLD 4006, Australia

**Keywords:** ataxia-telangiectasia, ataxia-telangiectasia mutated protein kinase, metabolic stress, mitochondrial dysfunction, anaplerosis

## Abstract

The ataxia-telangiectasia mutated (ATM) protein kinase is, as the name implies, mutated in the human genetic disorder ataxia-telangiectasia (A-T). This protein has its “finger in many pies”, being responsible for the phosphorylation of many thousands of proteins in different signaling pathways in its role in protecting the cell against a variety of different forms of stress that threaten to perturb cellular homeostasis. The classical role of ATM is the protection against DNA damage, but it is evident that it also plays a key role in maintaining cell homeostasis in the face of oxidative and other forms of non-DNA damaging stress. The presence of ATM is not only in the nucleus to cope with damage to DNA, but also in association with other organelles in the cytoplasm, which suggests a greater protective role. This review attempts to address this greater role of ATM in protecting the cell against both external and endogenous damage.

## 1. Introduction

Ataxia-telangiectasia (A-T) is an autosomal recessive disorder with features that include immunodeficiency, lung disease, radiosensitivity, cancer susceptibility and neurodegeneration [1,2]. Mutations in the Ataxia-telangiectasia mutated (*ATM*) gene that codes for a phosphoinositide-3-kinase-like kinase, ATM, are responsible for A-T [3]. The best described stimulus for ATM activation is the presence of DNA damage, primarily DNA DSB [4]. This agrees well with the hypersensitivity of ATM-deficient cells to ionizing radiation, a hallmark of A-T [5,6]. The activation of ATM by DNA DSBs leads to the phosphorylation of thousands of substrates that participate in the control of the cell cycle, DNA DSB repair, apoptosis, transcription and other cellular processes [7]. It seems likely that the alteration to the super-helical structure of chromatin by the introduction of a DSB in DNA leads to a low level of activation of ATM, followed by its recruitment to the sites of the DSB by the MRE11/Rad50/NBS1(MRN) complex and consequent amplification of the activation [8,9]. Autophosphorylation at several sites plays a key role in this process leading to the conversion of an inactive dimer into an active monomer [10]. Once activated, ATM phosphorylates all three members of the MRN complex, which, in turn, mediate the phosphorylation of multiple downstream substrates [11] (Figure 1).

ATM is also activated by oxidative and other forms of stress by a distinct mechanism that involves the formation of a disulfide cross-linked dimer [12]. The autophosphorylation of ATM on Ser 1981 and the phosphorylation of its substrates, p53 and Chk2 (not stably associated with DNA), occur in response to H_2_O_2_, but the phosphorylation of H2AX, a marker of DNA DSB and Kap1, a heterochromatin protein, both associated more strongly with chromatin, is not observed after H_2_O_2_, indicating that this response occurs in the absence of DNA damage (see Figure 2).

Evidence was provided that Cys^2991^ is involved in disulfide bond formation. Wild-type homo-dimeric ATM showed a normal response to MRN and DNA in vitro, but was not activated by H_2_O_2_. These results suggest that ATM acts as a redox sensor and, again, similar to DNA damage, has the capacity to phosphorylate a series of downstream substrates in response to oxidative stress. The phosphoproteomic analysis of ATM-dependent substrate phosphorylation is much more limited than that following DNA damage [13,14]. Cytoplasmic proteins were among those identified, as might be expected for ATM responding to stimuli outside the nucleus. However, the number of candidates were considerably less than those that were phosphorylated in response to DNA damage [15]. In normal conditions, mitochondria produce reactive oxygen species (ROS), which participate in intracellular signaling [16]. On the other hand, a general feature of A-T cells is the elevated basal levels of ROS, which is thought to arise by mitochondrial dysfunction [17]. The observation that mitochondrial-derived H_2_O_2_ produces ATM dimerization that is localized to the nucleus, suggests that ROS-mediated stress signaling relays mitochondria to the nucleus [18]. One function of ATM is the promotion of a glucose flux through the pentose phosphate pathway producing NADH for a robust antioxidant response. It has been proposed that the loss of the mitochondrial ROS-sensing function may be responsible for ROS accumulation and oxidative stress in A-T. These results suggest that, while ATM is activated by both DNA damage and non-DNA damage mechanisms, there is an integration of the response to different forms of cellular stress to maintain cellular homeostasis. Studies in yeast show that multiple genes involved in the DNA damage response (DDR) are partially or fully shared in common types of synthetic lethality, suggesting a specific cross-talk between DDR and the ROS stress response [19]. It is also evident that other forms of stress (e.g., metabolic stress) activate ATM, and it is likely that at least part of that is mediated by ROS [20].

The structural changes associated with ATM activation are discussed in detail in a recent review [21]. In short, ATM activation is controlled by the dimer interface of the ATM molecules that prevents accessibility where the active sites are shielded by a framework of HEAT repeats [22]. It forms a tightly packed closed dimer with an interface that is larger than a second form: an open dimer with a limited intermolecular interface. The open dimer contains an active site that is compatible with substrate binding, whereas a disordered PRD region blocks the substrate peptide from entering the active site in the closed dimer. The autophosphorylation, acetylation and Cys^2991^ sites for the activation of ATM in response to different stimuli all occur in disordered loops [22]. It seems likely that the two states are in equilibrium and that these post-translational forms drive the equilibrium in favor of the open form, which allow substrate binding and monomerization, perhaps simultaneously.

## 2. ATM Is Present in the Nucleus and Associated with Organelles in the Cytoplasm

In order to understand the overall role of ATM in protecting the cell, it is important to appreciate that this protein is present in both the nucleus and cytoplasm [23]. This is schematically depicted in Figure 3.

Subcellular fractionation, immune electron microscopy and immunofluorescence revealed that a small proportion (<10%) of the ATM protein was cytoplasmic in proliferating cells. This was subsequently localized to peroxisomes and shown to be involved in the import of proteins to the peroxisomal matrix [24]. The association with B-Adaptin indicated a role in intracellular vesicle transport [25]. On the other hand, ATM was detected exclusively in the cytoplasm of human Purkinje cells, supporting a major role for this protein in protecting neurons from oxidative stress [26]. In agreement with this finding, ATM was also predominantly present in the cytoplasm of mouse Purkinje cells and, in its absence, there was an increase in lysosomal numbers, which is discussed later in the study, in relation to mitophagy [27]. During its development in *Xenopus*, increased levels of ATM were detected in the nuclei of developing somites and in the central nervous system [28]. It has also been shown that a fraction of ATM proteins is localized in mitochondria, and mitochondrial damage can directly activate ATM kinase in the absence of DNA damage signaling [29]. Similar results were obtained by Morita et al., who showed that ATM could be activated in response to the oxidative stress in a mitochondrial-enriched subcellular fraction in a DNA damage response-independent fashion [30]. A more specific localization of ATM was demonstrated by Blignaut et al., who showed that ATM is found endogenously within cardiac myocyte mitochondria, under normoxic conditions, associated with the inner mitochondrial membrane and the electron transport chain [31]. However, to complicate this, it has been shown that mitochondrial-derived hydrogen peroxide promoted DNA dimerization, but the dimers were localized to the nucleus, suggesting a stress-signaling relay from the mitochondria to the nucleus [18]. ATM redox sensing promoted a glucose flux through the pentose phosphate pathway (PPP), thus increasing the abundance of glucose-6-phosphate dehydrogenase (G6PD) and cellular antioxidant capacity. It has been proposed that the loss of the mitochondrial ROS-signaling function of ATM may cause cellular ROS accumulation and oxidative stress in A-T cells (see Figure 2). As indicated above, it is evident that ATM is present in both the nucleus and cytoplasm associated with peroxisomes, mitochondria and vesicles. In proliferating cells, it participates in the DNA damage response to DNA DSBs, primarily in the nucleus. Earlier research points to a major component of ATM in the cytoplasm in neurons, but there is some disagreement on this. In responding to mitochondrial ROS, it makes sense that ATM would be associated with that organelle. Whether that might be an association with the organelle or localized to the inner membrane remains unclear, and no substrates for this mitochondrial kinase have been identified. The picture is more defined for peroxisomal ATM, where it is localized by its interaction with the PEX5 peroxisome importer receptor [24]. ATM phosphorylates PEX5 (ser 141) in response to ROS, which promotes PEX5 monoubiquitylation (Lys 209). This form of PEX5 is recognized by the autophagy adaptor protein p62, directing the autophagosome to the peroxisomes to induce selective autophagy, termed pexogaphy [32]. It is noted that this may be necessary, but not sufficient to induce pexophagy. Thus, A-T cells are defective in both pexogaphy and mitophagy, as discussed below, under mitochondrial dysfunction (Figure 4).

ATM is activated by DNA DSB and by oxidative and other forms of stress, and the observation that mitochondrial-induced H_2_O_2_ promotes DNA dimers that appear in the nucleus suggest some form of a signaling between these organelles [18]. It is unclear why ATM, activated outside the nucleus in the absence of DNA damage, should migrate to the nucleus, unless it is phosphorylating substrates there. This supports a more general and coordinated role for ATM in responding to stress, whether activated by DNA DSB or by ROS. Overall, its role is to protect the cell on more than one level and, in so doing, it would be beneficial if this was in a coordinated way. For example, in a proliferating cell, ATM might be expected to be predominantly nuclear to protect replicating DNA, which it is, but in differentiated neuronal cells that are more cytoplasmic, which is again supported by most observations.

## 3. Impaired Intracellular Signaling in A-T

A recent report suggests that impaired endoplasmic reticulum–mitochondrial signaling may account for the mitochondrial dysfunction in A-T [20]. This raises the issue of a more general signaling defect in these cells. A reduced lifespan and early senescence of A-T fibroblasts were shown to be rescued by the inclusion of growth factors in media [33]. Many years on, it is now evident that a number of factors in the A-T cell phenotype contribute to the premature aging observed in A-T cells. These include an impaired DNA damage response, telomere shortening, oxidative damage and mitochondrial dysfunction, all of which depend on intracellular signaling. Evidence of a defect in signal transduction in A-T has been provided in several reports. O’Connor and Linthicum demonstrated that A-T lymphocytes were defective in the transmission of a mitogen-mediated signal from the cytoplasm to the nucleus, or due to an inability of the nucleus to respond [34]. The morphological features of blastogenesis were totally absent in phytohemagglutinin (PHA)-stimulated lymphocytes. The intracellular mobilization of Ca^2+^ in T cells from A-T patients, in response to PHA and the anti-CD3 antibody, was reported, by Kondo et al., to be defective [35]. These data pointed to a defect in signal transduction through the CD3 complex on the surface of these cells. A more detailed study also provided evidence for defective signaling through the B-cell antigen receptor in A-T lymphoblastoid cells [36]. The cross-linking of the B-cell receptor (BCR) led to a defective mitogen response in these cells, and Ca^2+^ mobilization from internal stores was either absent or greatly reduced. In line with the defect in Ca^2+^ mobilization, PLCγ activation was reduced in one A-T cell line and negligible in two others, indicative of a defect of one arm of the cascade activated by BCR cross-linking [37]. Furthermore, when levels of IP3, resulting from a degradation of phosphoinositol 4,5 bisphosphate (PIP2) by PLCγ, were determined, A-T cells failed to show an increase. However, Speck et al. 2002 failed to reproduce these results in other B-lymphoblastoid cells [38]. Nevertheless, these observations raise the possibility that the hypo-responsiveness of the transmembrane signaling, primarily observed in T lymphocytes and also in B cells from A-T patients, accounts for, at least in part, immune dysfunction, which is also a characteristic of the response of the humoral arm with this syndrome. The loss of ATM kinase also results in the up-regulation of the anti-apoptotic inhibiting protein, FLIP, which competes with pro-caspases to bind to the death-inducing signaling complex and render cells significantly more resistant to FAS- and TRAIL-induced apoptosis [39]. This suggests that ATM loss may sustain lymphoma development as a consequence of the resistance to FAS-induced apoptosis resistance. This is in conflict with the more widely held view that tumor development in A-T is due to a defective response to DNA damage.

Impaired signaling between the ER and mitochondria was recently described in relation to A-T cells in response to nutrient deprivation (glycolysis inhibition) (Figure 5) [20].

The disruption of the ATM gene in the human bronchial epithelial cell line (HBEC) using CRISPR-Cas9, sensitized the resulting cells to nutrient deprivation to an extent that was three-fold greater than in wild-type cells, and this was similar to the hypersensitivity observed in A-T cells to ionizing radiation or oxidative stress. This sensitivity to nutrient deprivation was widely observed in A-T cells, including primary airway epithelial cells, induced pluripotent stem cells (iPSCs) and olfactory neural stem (ONS) cells [20].

## 4. ATM and Inflammation

Early studies on the pivotal role of nitric oxide in the induction of cellular stress and the activation of a p53 response pathway, implicated ATM activation during chronic inflammation [40]. ATM is activated by ROS, which is also elevated in ATM-deficient cells and produces inflammatory changes. A connection between inflammatory stimuli that trigger the activation of the NF-kB response linking DNA damage response to this response, mediated by ATM, has been described [41] (McCool and Miyamoto 2012). The activation of NF-kB regulates the expression of genes involved in the response to oxidative stress and inflammatory changes, and thus, the loss of ATM might be expected to have an impact on this expression. Indeed, genotoxic stress leads to the ATM-dependent phosphorylation of p65(RelA) that represses the transcription of specific genes [42].

There is evidence that systemic inflammation and oxidative stress contribute to the pathophysiology of lung disease in A-T [43]. Serum levels of the pro-inflammatory cytokines, IL-8 and IL-6, were shown to be elevated in patients with A-T, suggesting that markers of systemic inflammation may be useful in identifying the individuals with A-T at an increased risk of lower lung functions [43]. Airway epithelial cells from patients with A-T are also characterized by elevated levels of the pro-inflammatory cytokines IL-8 and TNF-α, following infection with S. pneumoniae, which would also support the role of inflammation in the process. Other evidence for an inflammatory phenotype in A-T patients was provided by Hartlova et al. (2015), who showed that unrepaired DNA damage released into the cytoplasm in *Atm*^−/−^ mice primed the type 1 interferon response through the STING pathway to promote anti-microbial immunity [44]. The use of olfactory neurosphere-derived cells and brain organoids also revealed that increased cGAS and STING activity was an important contributor to chronic inflammation in the central nervous system in A-T [45]. It was subsequently showed that in the absence of ATM, the defect was in inflammasome-dependent anti-bacterial innate immunity. Diminished interleukin-1B (IL-1B) production, in response to bacteria, was observed in patients and *Atm*^−/−^ mice. Their data suggested the negative regulation of inflammasome formation by ROS, which could account for the susceptibility of patients to pulmonary bacterial infection. This is keeping with the broader observation that ROS is elevated in all A-T cell types investigated to date. Thus, in addition to the potential of increased ROS to cause DNA and more general cellular damage, these results point to an additional mechanism of interfering with the immune response, and thus increased susceptibility to infection. A role for oxidative damage in impaired innate immunity was also observed in airway epithelial cells from A-T patients [43]. They demonstrated that A-T epithelial cells were extremely sensitive to oxidative stress induced by H_2_O_2,_ and hypothesized that the increased susceptibility of patients to respiratory infection could be explained by the ability of microorganisms, such as Streptococcus pneumoniae, to produce H_2_O_2_ as the damaging agent. They showed that a heightened susceptibility to infection could be explained by both increased oxidative damage and a defect in inflammasome activation. A defect in the ASC–caspase 1 signaling pathway and decreased levels of the inflammasome-dependent IL-1B were observed in patient cells.

## 5. Mitochondrial Dysfunction in A-T

A mitochondrial disease database was established, which outlines the clinical features observed in these mitochondrial diseases (www.mitodb.com; accessed on 11 February 2022) and allows for the comparison of symptoms in other disorders to determine whether they may fall into a mitochondrial phenotype. Based on this database [46], extensive bioinformatic tools were developed, which revealed that A-T could be characterized as a mitochondrial disorder [46]. The scoring system over a range 0–100 placed A-T at 90 when neurological involvement was included. They suggested that disorders of this nature would thus be prime targets for further investigation, and potentially for treatment with drugs known to augment mitochondrial function. In keeping with this prediction, it is not surprising that there exists an accumulating body of evidence for mitochondrial dysfunction in A-T [47]. The state of continuous oxidative stress and the constitutive activation of pathways that normally respond to oxidative damage, led Gatti and his colleagues to determine whether the oxidative stress phenotype of A-T cells might reflect an intrinsic mitochondrial dysfunction [48]. They observed a sub-population of mitochondria with lower membrane potential in A-T cells, the basal expression of mitochondrial DNA repair and ROS scavenging genes were elevated, mitochondrial respiration and oxidation rates were greatly compromised, and the latter was partially corrected in response to ATM restoration in A-T cells. They concluded that at least some of the oxidative stress observed in A-T cells could be attributed to intrinsic mitochondrial dysfunction. Intrinsic mitochondrial abnormalities in *Atm*^−/−^ mouse thymocytes were later reported by Valentin-Vega et al., including elevated reactive oxygen species, increased mitochondria mass, a high cellular respiratory capacity, and decreased mitophagy [29]. It was subsequently shown that spermidine triggers PINK1/Parkin-mediated mitophagy in control cells, but not in ATM-deficient cells [49]. Studies in *C. elegans* and *Atm*-deficient mice showed that the loss of ATM induced the accumulation of damaged mitochondria, mitochondrial dysfunction and compromised mitophagy due to NAD+ insufficiency [50]. They showed that replenishing intracellular NAD+ improved mitochondrial quality by increasing mitophagy. Furthermore, it reduced the severity of neuropathology and extended the lifespan of both animal models. The conclusion from these data was that both the accumulation of DNA damage and mitochondrial dysfunction contribute to the pathophysiology of premature aging in A-T.

The role of ATM in more general autophagy is somewhat confusing in that it has been reported to be impaired, and both upregulated and downregulated in ATM-deficient cells [29,51,52]. For example, it has been reported that autophagic flux is upregulated in *Atm*^−/−^ lysosomes associated with a more acidic pH. In short, when ATM is missing or inhibited, autophagy levels increase, which appears to be the opposite to the defect in mitophagy in several reports (see below). Valentin-Vega et al. presented some evidence that autophagy was upregulated in ATM-null thymocytes and MEFs, showing a marked reduction in p62/sequestosome and an increased ratio of LC3II/LC3I [29]. However, this is evidence for a defect in mitophagy, with striking increases in the number of altered mitochondria and the mitochondrial mass in *Atm*^−/−^ mouse thymocytes [29]. It is notable that the mitochondrial DNA content varied in different ATM-deficient cells and was noticeably increased only in early passage fibroblasts. It is not clear why this is the case, since increased mitochondrial content would suggest increased mtDNA. The increased mitochondrial content did not appear to be as a result of increased biogenesis, but rather decreased mitophagy. However, the evidence for decreased mitophagy was not clear-cut from Parkin levels, higher basal levels or the lack of change as a result of CCCP treatment, but it was evident that the loss of COX IV was deficient in A-T fibroblasts. It was subsequently shown that spermidine triggers PINK1/Parkin-mediated mitophagy in control cells, but not in ATM-deficient cells [48]. In addition, the formation of mitophagosomes and mitolysosomes, as well as a decrease in mitochondrial mass, were shown to be ATM dependent. Studies in *C. elegans* and *Atm*-deficient mice showed a loss of the ATM-induced accumulation of damaged mitochondria, mitochondrial dysfunction and compromised mitophagy due to NAD+ insufficiency [49]. They showed that replenishing intracellular NAD+ improved mitochondrial quality by increasing mitophagy [50]. This was achieved using a mitochondrial reporter strain to visualize the expression and colocalization of LGG-1 and DCT-1 in Atm worms. Furthermore, it reduced the severity of neuropathology and extended the lifespan of both animal models [51]. A significant increase in mitochondrial mass and mitophagy at 50% of that in the controls was more recently observed in ATM-deficient human bronchial epithelial cells after nutrient deprivation [20]. A defect in mitophagy and the increased mitochondrial content reported for A-T cells would be expected to create an imbalance in the copy number and the persistence of damaged mitochondria that would interfere with normal mitochondrial function. However, in a recent report, ionophore-induced mitochondrial damage and mitophagy triggered ATM activation through the generation of the ROS function [52]. While antioxidants inhibited ROS production and ATM activation, but failed to prevent mitophagy, this suggested this form of mitophagy does not require ATM [53]. Indeed, the same group showed that FCCP-induced mitophagy in cancer cell lines was dependent on ATM, but not on its kinase, activity [54]. This would also have an impact on the mtDNA copy number and perhaps its integrity, suggesting a possible role for ATM in the response to DNA damage in mitochondria. The elevated mitochondrial ROS, which appears to be a characteristic of all A-T cell types, would be expected to damage mtDNA [47]. Indeed, there is evidence for a four-fold increase in mtDNA damage in A-T fibroblasts and an impaired capacity to remove oxidative lesions [55]. However, there was some variability between the control fibroblasts in the total lesions and extent of repair. The impaired capacity can be explained by reduced levels of ligase III in mitochondria, in the absence of ATM. This is the only DNA ligase in mitochondria and is required to maintain the integrity of mtDNA. In its absence in mice, severe ataxia is observed [56]. This is the only report of a defect in mitochondrial DNA damage and its repair and needs to be viewed with some caution.

Since Valentin-Vega et al. provided evidence that autophagy was upregulated in ATM-null cells by the allelic loss of the autophagic regulatory gene Beclin-1, they investigated its effect on tumorigenesis [29]. While the development of T-cell lymphomas and Beclin-1 heterozygosity in *Atm*^−/−^ mice is also associated with tumor development, Valentin-Vega et al. unexpectedly observed that the allelic loss of *Beclin-1* in *Atm*^−/−^ mice increased survival to 262 days compared to 137 days in *Atm*^−/−^ mice. Furthermore, the delayed tumor development in *Atm*-null mice associated with *Beclin-1* heterozygosity that was correlated with the rescue of mitochondrial defects, not the rescue of DDR abnormalities. However, they could not exclude the possibility that *Beclin-1* heterozygosity might also delay tumor development by compromising the survival of developing malignant cells. Nevertheless, *Beclin-1* heterozygosity appears to modulate mitochondrial homeostasis, which has an impact on the lifespan of *Atm*^−/−^ mice. A greater insight into the ATM–Beclin 1 interaction was provided in the demonstration that an ATM/Chk2/Beclin-1 axis protected cells from oxidative stress by promoting autophagy [56]. They demonstrated that Chk2 mediated autophagy by the phosphorylation of Beclin-1, by limiting ROS levels during nutrient deprivation. Under these conditions, by sensing ROS, the ATM/Chk2/Beclin-1 complex would control the levels of ROS, clear damaged mitochondria and prevent cell death. The various abnormalities in mitochondria led Valentin-Vega et al., [27] to suggest that A-T should be considered, at least in part, as a mitochondrial disease.

Since oxidative stress and mitochondrial dysfunction are implicated in A-T [47], it was discussed whether reducing mitochondrial reactive oxygen species (ROS) by overexpressing catalase targeted to mitochondria (mCAT) might alleviate the A-T-related pathology in *Atm*^−/−^ mice [57]. This approach led to a number of beneficial effects in *Atm*^−/−^ mice, including a reduced propensity to develop thymic lymphomas, improved bone marrow hematopoiesis and macrophage differentiation in vitro, and the partial rescue of memory T-cell developmental defects. These data support a role for mitochondrial ROS in A-T-related pathology, and the authors raise the possibility that antioxidant therapies directed at mitochondrial ROS may be of therapeutic value for A-T patients. This approach has the benefit, over the more generalized use of antioxidants, of alleviating the A-T phenotype; however, evidently, being able to deliver a construct expressing CAT in the appropriate region or cells in the brains of patients represents a major hurdle. However, what emerges from this report is that being able to improve mitochondrial function by reducing the toxic effects of ROS has the potential to address the pathology in patients. This is covered in more detail when an anaplerotic approach to correcting mitochondrial function is discussed later. While this defect may be downstream of the defect in ATM function, it may nevertheless be a useful target for the treatment of patients.

## 6. Metabolic Stress and A-T

A number of A-T features, including insulin resistance, are difficult to explain via a defect in the response to DNA damage. It seems likely that ATM-dependent stress pathways mediate susceptibility to metabolic syndrome, since chloroquine, which promotes ATM activity and inhibits the JNK stress kinase, increases sensitivity to insulin and decreases vascular disease [58]. Recently, an A-T patient cohort study demonstrated that diabetes is common, especially in older A-T patients, and often begins at puberty [59]. ATM deficiency is associated with insulin resistance and diabetes [60]. Specifically, an early study showed that the insulin-dependent dissociation of 4E-BP1 is significantly reduced in ATM-deficient cells [61]. The study went on to show that ATM phosphorylates 4E-BP1 at Ser 111 and that both in vitro and in vivo treatments with insulin induce this phosphorylation in an ATM-dependent manner, confirming the contribution of ATM to the metabolic abnormalities in A-T. Further, animal studies demonstrated that loss of one or both alleles of ATM increases the symptoms of metabolic syndrome, including glucose intolerance and insulin resistance in ApoE^−/−^ mice fed on a high-fat diet [62]. They showed that during insulin resistance induced by ATM deficiency, JNK was activated, which consequently increased the level of inhibitory serine-307 phosphorylation on the insulin receptor substrate-1 (IRS-1) and caused insulin resistance. Other results suggest that lower ATM levels in a rat model fed a high-fat diet may contribute to the development of insulin resistance by downregulating Akt activity, since defective Akt activation is an important mechanism in the development of this resistance [60]. They also showed that in cells transfected with wild-type ATM, insulin caused a dramatic increase in the cell surface glucose transporter 4 (GLUT4), while in cells transfected with kinase-dead ATM, the translocation of GLUT4 to the cell surface in response to insulin was markedly inhibited. Therefore, the reduced PI3K/Akt signaling during ATM deficiency appears to contribute to the decrease in GLUT4 translocation and insulin resistance. However, it is evident that several pathways are involved in this signaling and the role of ATM is more complex. While these results offer a greater insight into metabolic syndrome, they raise the possibility that ATM has a greater role in regulating metabolism. A better understanding of this was obtained when it was shown that mice possessing a mutation in an ATM target site on p53 (S18A) showed increased metabolic stress, including increased inflammatory cytokines, reduced antioxidant gene expression and defects in glucose homeostasis, such as insulin resistance [62]. They also showed that deregulated ROS levels contributed to an imbalance in glucose homeostasis. As mentioned above, enhanced levels of ROS are a characteristic of A-T cells, suggesting that the well-established oxidative stress phenotypes of these cells, at least partially, contribute to metabolic stress [62].

It is evident that the efficient generation of ATP through glycolysis and mitochondrial oxidative phosphorylation is critical to cell survival. This is particularly acute for the survival of neurons, where the restoration of membrane potential after spiking is heavily ATP dependent [63]. Given the accumulating reports on mitochondrial dysfunction in a variety of A-T cell types, it might be predicted that this would have an impact on neurons in the CNS of A-T patients. Gene expression analysis in the cerebellar cortex predicted a generalized loss of nuclear-encoded mitochondrial proteins in A-T, emphasizing the importance of ATM in protecting mitochondrial function [63]. These data suggested that the demand for ATP might be more acute in the cerebellum of patients with A-T. They showed that the depletion of ATP generated ROS, which activated ATM and, in turn, led to the phosphorylation of nuclear respiratory factor 1 (NRF1) [64]. NRF1 was dimerized, translocated to the nucleus where it upregulated nuclear-encoded mitochondrial genes, enhancing the capacity of the electron transport genes and restoration of mitochondrial function. The replenishment of ATP was defective in Atm-deficient cells and chronic ATP depletion impacted on cell survival. The implication was that cerebellar cells would be exposed to an ATP deficit, which would contribute to their degeneration.

## 7. Endoplasmic Reticulum (ER) Stress

The unfolded protein response (UPR) is induced by the presence of unfolded proteins by the activation of membrane sensors, leading to the downregulation of protein synthesis and acceleration of protein secretion [65]. This occurs in the endoplasmic reticulum (ER) where protein aggregation causes inflammation and ER stress, both of which are observed in A-T cells. The deregulation of proteostasis may be a common feature in neurodegeneration [66]. ER quality control (ERQC) regulates ER homeostasis by three mechanisms: ER-associated degradation (ERAD), the unfolded protein response (UPR) and autophagy [66]. The UPR is achieved through the activation of three transmembrane sensors on the ER, IRE1a, PERK and ATF6, and the upregulation of genes downregulating protein synthesis. These processes are initiated by ER stress in response to the accumulation of unfolded proteins in the lumen of the ER to rescue the cell. Yan et al. were the first to provide evidence for oxidative damage leading to ER stress in *Atm*^−/−^ thymocytes and some evidence for UPR (peIF2α) [67]. They proposed that ER stress and the unfolded protein response are secondary to oxidative stress in *Atm*^−/−^ thymocytes. On the other hand, Poletto et al. showed that persistent oxidative stress in ATM-depleted fibroblasts, which was buffered by adjustments to proteostasis, countered protein damage due to this stress [68]. A greater insight into this defect was obtained by employing the separation of function mutations for ATM activation by DNA damage and oxidative stress in a Tet-inducible system in U2OS cells [69]. They showed that that a variant incapable of activation by oxidative stress resulted in a global increase in protein aggregation. Gene enrichment analysis revealed that there was a strong enrichment of nuclear proteins, including the RFC complex, Mre11 and Rad50 proteins, and RNA processing enzymes. The same group showed that this protein aggregation was dependent on Poly-ADP-ribosylation activity (PARP) [70]. The PARP inhibitor, veliparib, reduced CK2B pSMB2 aggregation and the depletion of PARP1 also reduced aggregation. Their data suggest that the hyper-PARylation after ATM loss occurs in the vicinity of single strand breaks and R-Loops after nuclear damage in these cells. The analysis of cerebellum lysates from A-T patients showed significantly increased levels of aggregates for IP3R1, CA8, INPP5A and CBLN1, important for inositol-regulated Ca^2+^ signaling. In addition, IP3R1 deficiency causes familial spinocerebellar ataxia.

As mentioned previously, Ca^2+^ signaling abnormalities were already reported in ATM-deficient cells. Yan et al. demonstrated defective signal transduction through the activation of protein tyrosine kinase p53/p56lyn in A-T lymphoblastoid cells [71]. While Ca^2+^ was mobilized largely from internal stores post-irradiation in control cells, this mobilization was either absent or increased very slowly in A-T cells [72]. These data point to ATM-dependent radiation-induced plasma membrane/cytoplasmic signaling that leads to the mobilization of Ca^2+^. In this case, the stressor was radiation, but, as suggested by the authors, it may reflect a more general signaling defect. It was speculated that the defect is likely to be due to the reduced production of inositol 1,4,5 trisphosphate (IP3) by the defective activation of PLCγ in the signaling pathway. This has parallels to BCR cross-linking-induced signaling in A-T cells, which identified a defect in PLCγ activation and IP3 production [36]. More recently, evidence of the reduced interaction between ER and mitochondria in ATM-deficient cells in response to metabolic stress has been reported [20]. The prediction was that there would be a defect in Ca^2+^ release from the ER and transfer to mitochondria in these cells under these conditions. As predicted, a significantly reduced release of Ca^2+^ from the ER and reduced uptake into mitochondria was observed under the conditions of nutrient deprivation (glycolysis inhibition) (Figure 5). It was also shown that this affects mitochondrial function and cell survival because the inhibition of the IP3R and consequently Ca^2+^ release and transfer in control cells led to significantly increased killing, similar to that observed for ATM-deficient cells treated with 2DG (inhibitor of glycolysis) alone. Consistent with the defect in Ca^2+^ homeostasis in ATM-deficient cells, these cells failed to efficiently assemble the IP3R1-GRP75-VDAC1 complex, which has been shown to facilitate the release of Ca^2+^ from the ER and uptake into mitochondria [72]. That study did not investigate events upstream of the IP3R1, but, in line with the defect in BCR signaling, it might be expected that the generation of IP3 would be defective and, as a consequence, result in its interaction with IP3R1 for the release of Ca^2+^ stores from the ER in response to nutrient stress.

## 8. Focusing on a Therapy for A-T

To date, there is no cure for A-T, but certain characteristics of the disorder are amenable to therapy. Supplemental gamma globulins are prescribed for patients with hypogammaglobulinemia; sinopulmonary infections are treated with antibiotics and antioxidants and anti-inflammatory agents have also been employed in treatments, but none of these have been successful in significantly slowing down the progress of the disorder or curing the disease [73]. This a progressive movement disorder characterized by a deterioration of the neurologic function, a decline in pulmonary function and the development of hematopoietic malignancies. As a consequence, a reduced quality of life occurs, and the overall life expectancy is markedly reduced. An excellent review on the current treatments and new emerging therapies was published by Pinho de Oliveira et al., [74] so the aim of the present study is not to repeat the various approaches already described, but rather to focus on oxidative stress and how reducing this may present possible approaches to therapy. A characteristic that appears to be universal in A-T cells is higher basal levels of ROS and oxidative stress. Not only is ROS elevated in A-T cells, but these cells are also hypersensitive to oxidative stress. Yi et al. showed higher levels of micronuclei in A-T cultures exposed to H_2_O_2_ [75] and a five-fold increase in cell killing in A-T primary airway epithelial cells after the exposure to different concentrations of H_2_O_2_ was reported [76]. A heightened susceptibility to streptococcal infection due to increased oxidative damage was also observed in epithelial cells from patients with A-T [43]. Metabolic disorders implicated in cardiovascular and liver disease are frequently observed with increasing age in A-T patients, and it has been suggested that these may be treated with nutritional intervention and the use of antioxidant drugs [77]. Oxidative stress has also been associated with non-alcoholic steatohepatitis in a patient with A-T [78]. In addition, the levels of superoxide dismutase and catalase were elevated in the erythrocytes of A-T patients an indicator of oxidative stress [79]. There is also evidence that antioxidant capacity is reduced in the serum of A-T patients [80]. To determine whether some of the abnormalities observed in A-T patients might be due to oxidative stress, a clinical trial was conducted at the A-T Clinical Center John’s Hopkins, using α-lipoic acid and nicotinamide, but while some improvements in the levels of markers associated with oxidative stress were observed, this approach was not successful overall. There are several reports with Atm-deficient animal models that antioxidants improve the neurological phenotype, suggesting that neutralizing ROS can have a beneficial effect. For example, oxidative stress is responsible for deficient survival and dendritogenesis in Purkinje neurons in *Atm*-deficient mice, which is corrected by isoindoline nitroxide, CTMIO [81]. Another approach would be to prevent excess ROS arising in the first instance by improving mitochondrial function in A-T, for which there is considerable evidence of mitochondrial dysfunction in A-T cells [29]. The replenishment of NAD+ in both mouse and worm models of A-T-stimulated neuronal DNA repair and improved mitochondrial quality through mitophagy [82]. This treatment normalized TCA cycle intermediates, which were found to be lower in *Atm^−/−^* worms and *Atm*-deficient mice, improved mitochondrial function and extended the lifespans for both models. Thus, it is likely that improving mitochondrial function and reducing persistent damage in Atm-deficient cells both contribute to the improved phenotype in these animals. A clinical trial is underway to investigate the effect of dietary supplementation of nicotinamide riboside (NR) in children with A-T, with the main focus on the improvement of neurological symptoms (Hilde Nilsen, University Hospital, Akershus). In addition, liver function, blood sugar control and mitochondrial function will also be monitored. Recent data have shown that after glycolysis inhibition, signaling between the ER and mitochondrion is defective and that it impacts on mitochondrial function, as is evident from an increase in the oxygen consumption rates and defective mitophagy [20]. These data, together with several reports on mitochondrial dysfunction in A-T cells and animal models, suggested that correcting this dysfunction might alleviate some or all of the cellular A-T phenotypes. Triheptanoin, a triglyceride composed of three odd-chain fatty acids (heptanoate, C7), was employed for the treatment of several human disorders, where there is a defect in the supply of energy from the intermediate of the tricarboxylic acid (TCA) cycle or an impairment of fatty acid oxidation, neurological diseases and those due to disturbed glucose acid cycle intermediates, or where fatty acid degradation is impaired [83]. This compound enters the mitochondria without the need for the carnitine transport shuttle [84]. Triheptanoin recently received its first regulatory approval for use in the U.S. as a source of calories and fatty acids for the treatment of pediatric and adult patients with molecularly confirmed long-chain fatty acid oxidation disorders [85]. The metabolism of triheptanoin occurs to heptanoate, which is capable of entering mitochondria and is further converted to two molecules of acetyl CoA and one molecule of propionyl by β-oxidation. Propionyl CoA is further metabolized to methylymalonyl CoA and succinyl CoA (Figure 6).

In this way, they function as anaplerotic agents to replenish TCA cycle intermediates and ATP generation (Figure 6). In addition, one cycle of β-oxidation also generates C5 fatty acids, which, through a conversion to C5 ketone bodies, are neuroprotective. A second cycle not only generates acetyl CoA, but also succinyl CoA, which is also anaplerotic. Triheptanoin has been shown to mitigate brain ATP depletion and mitochondrial dysfunction, including respiration and redox balance in a mouse model of Alzheimer’s disease, supporting the energy failure hypothesis for that disorder [86]. Since mitochondrial dysfunction and abnormalities in energy metabolism are frequently observed in A-T cells, they represent a potential target for an anaplerotic approach to correction. Yeo et al. described the correction of ER–mitochondrial signaling and mitochondrial function in ATM-deficient cells using heptanoate [17]. They confirmed a hypersensitivity to nutrient deprivation in an A-T bronchial epithelial cell line, and that heptanoate (C7) reduced the extent of cell killing to levels comparable to those in controls. Protection against cell killing was also observed in primary airway epithelial cells from A-T patients, as well as in glucose-deprived olfactory neurosphere (ONS)-derived cells from patients. C7 also reduced high basal levels of ROS in A-T cells and increased the number of ER-mitochondrial contact sites in response to glycolysis inhibition. Previous results provided evidence that reduced ER-mitochondrial contacts in response to nutrient deprivation could account for a defect in Ca^2+^ release from the ER and transfer to the mitochondria under these conditions [20]. When C7 was included in metabolically stressed ATM-deficient cells, a significant increase in ER-Ca^2+^ release to levels comparable to controls was observed. This was also the case for the transfer of Ca^2+^ to the mitochondria. Several parameters of mitochondrial dysfunction in ATM-deficient cells were also corrected by C7. Higher basal levels of lactate dehydrogenase (LDHA) and further increases in metabolic stress, together with increases in lactate, supported a greater reliance on aerobic glycolysis in ATM-deficient cells [17]. Compatible with the anaplerotic effects of C7, the inclusion of this compound dramatically reduced LDHA and lactate levels while improving mitochondrial function in A-T cells (Figure 6). Overall, these data reveal that C7 corrects all aspects of ER–mitochondrial signaling and dramatically reduces cell killing in response to metabolic stress in ATM-deficient cells.

While triheptanoin was not used in these cellular studies, since this compound would be metabolized to C7, it is likely that it has the potential to be successful as a novel therapy for patients with A-T. Only one recent report described the use of triheptanoin to treat a human mitochondrial abnormality, a mitochondrial malate dehydrogenase (MDH2D) deficiency [87]. MDH2D is caused by an interruption of the TCA cycle malate–aspartate shuttle, which results in severe early onset encephalopathy. C7 produced from triheptanoin enters the mitochondria and is degraded by β-oxidation to C2 and C3 that enter the TCA cycle to generate intramitochondrial NADH, which compensates for the inability of MDH2D to take up cytosolic NADH into the mitochondria. Triheptanoin was well tolerated, and the patient’s neurologic and biochemical phenotype improved.

A Phase II clinical trial is currently underway to investigate the effects of triheptanoin on patients. The trial, “Ataxia-Telangiectasis: Treating mitochondrial dysfunction with a novel form of anaplerosis (A-TC7)”, is sponsored by the University of Queensland and its collaborator, the National Health and Medical Research Council of Australia.

## 9. Conclusions and Future Perspectives

The genetic disorder A-T was first described almost 100 years ago, but, to date, there is no cure for this disorder. After the *ATM* gene was discovered 25 years ago, it became quickly evident that the enhanced sensitivity of patients and of cells from these patients to ionizing radiation could be explained by the role of the gene product involved (ATM) in the response to DNA damage. It was also evident that oxidative stress was a major characteristic associated with A-T cells. Cells from patients were as sensitive to oxidative stress as they were to radiation, and there are multiple studies in cell lines and Atm-deficient animal models that antioxidants relieved the phenotype of this form of stress. Importantly, oxidative stress was also capable of activating ATM by a mechanism distinct from that caused by DNA damaging agents. Accordingly, while a large part of the A-T phenotype can be explained by a defective DDR, it is evident that oxidative stress is also a contributing factor. This is closely linked to mitochondrial abnormalities, which are extensively described in A-T cells, and, when corrected, alleviate aspects of the phenotype in animal models and cells in culture. These mitochondrial abnormalities are also linked to the hypersensitivity of a variety of A-T cells to nutrient deprivation, supporting the mitochondrial defects and a greater reliance of A-T cells on aerobic glycolysis when under metabolic stress. The use of antioxidants leads to a partial improvement in the response of A-T cells to this form of stress, implicating at least some involvement of oxidative stress in the process. The more complete correction of the nutrient hypersensitivity phenotype by the anaplerotic agent heptanoate, demonstrates that improving mitochondrial function has the potential for therapy in patients with A-T. It is clear that this approach is but one of the many being used or contemplated in the treatment of A-T. These include antioxidants, NAD+ precursors, glucocorticoids, read-through compounds and antisense oligonucleotide gene therapy, in a specific case.

## Figures and Tables

**Figure 1 antioxidants-11-00653-f001:**
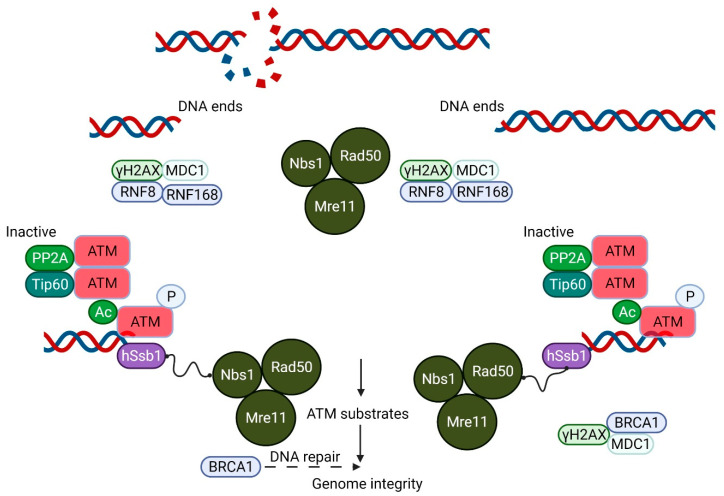
A Schematic model showing the essential role of ATM in DNA double-strand break (DSB) repair. The Mre11/Rad50/Nbs1 (MRN) complex plays an important role in DSB recognition and signaling ATM. ATM, which exists as an inactive dimer under normal conditions, becomes activated as a monomer in response to MRN signaling. This MRN-mediated activation of ATM stimulates the kinase activity of ATM, which phosphorylates p53, Chk2 and other substrates, including Brca1, which together play a crucial role in the maintenance of the genomic integrity of the cell.

**Figure 2 antioxidants-11-00653-f002:**
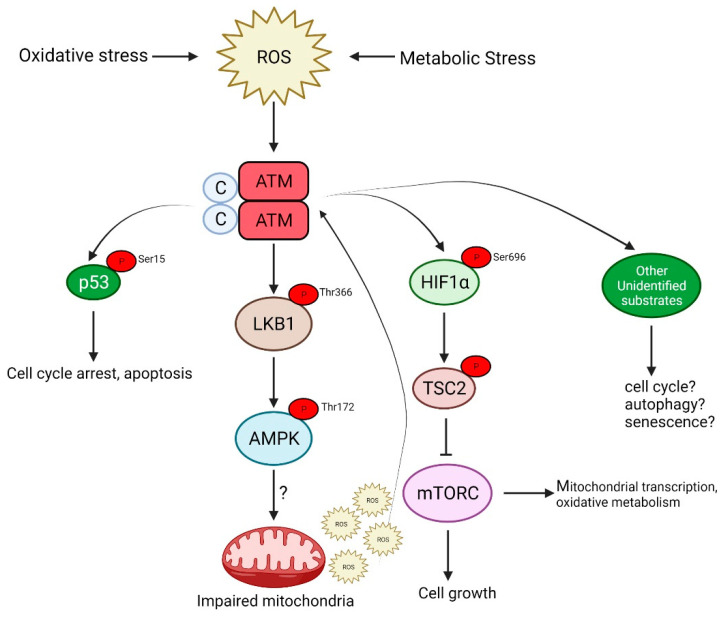
ATM is activated in response to oxidative stress caused by the increased production of reactive oxygen species (ROS). This activation is independent of the DNA damage signaling pathway and signals via AMPK, p53 or mTORC, which may culminate in either of the following: cell cycle arrest, autophagy and senescence or even apoptosis in the case of severe DNA damage.

**Figure 3 antioxidants-11-00653-f003:**
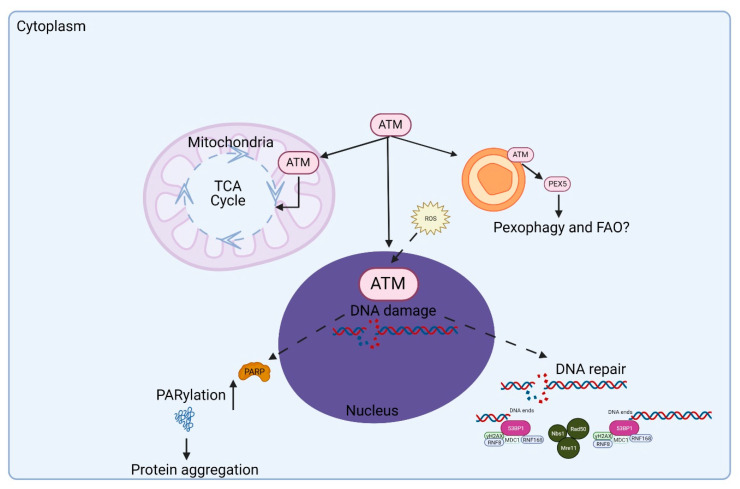
ATM is localized to various organelles within the cell, including the nucleus, peroxisomes, mitochondria and vesicles. In the nucleus, ATM is involved in DNA damage response and the regulation of transcription. Further, ATM is also present in the cytoplasm playing a variety of roles, including redox sensing and driving mitophagy, pexophagy and fatty acid β-oxidation in peroxisomes, proteostasis and calcium signaling in the vesicles and endoplasmic reticulum, respectively.

**Figure 4 antioxidants-11-00653-f004:**
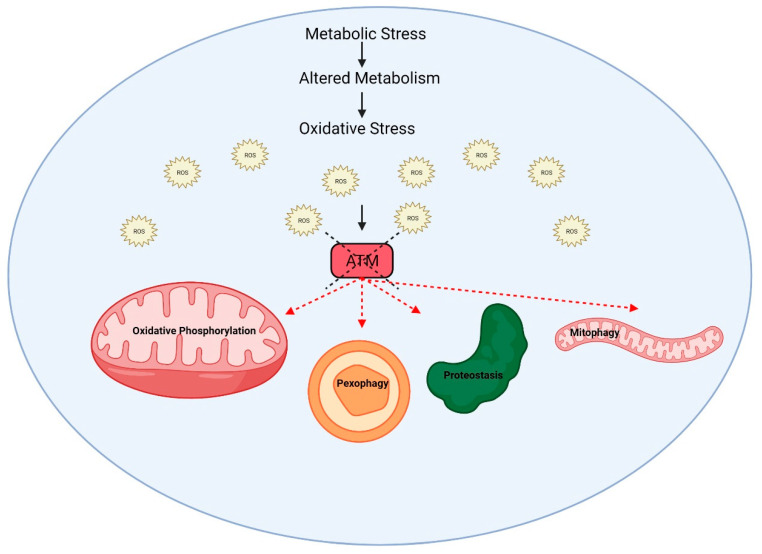
ATM has a broader role in regulating metabolism by modulating mitochondrial and peroxisomal enzymes involved in oxidative phosphorylation in fatty acid β- oxidation pathways. A-T cells lacking the ATM function show increased metabolic stress, a reduced expression of antioxidant genes, which cause a severe dysregulation of ROS and an imbalance in glucose homeostasis.

**Figure 5 antioxidants-11-00653-f005:**
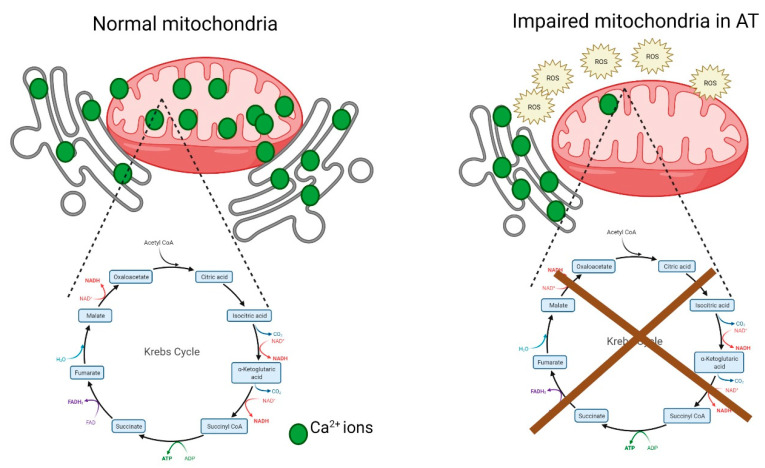
The endoplasmic reticulum (ER) mitochondria form contact sites that control lipid, calcium homeostasis and mitochondrial metabolism. The transfer of Ca^2+^ ions from the ER reserves to mitochondria is crucial for the TCA cycle, as Ca^2+^ ions act as catalysts for several enzymatic interconversions within the TCA cycle.

**Figure 6 antioxidants-11-00653-f006:**
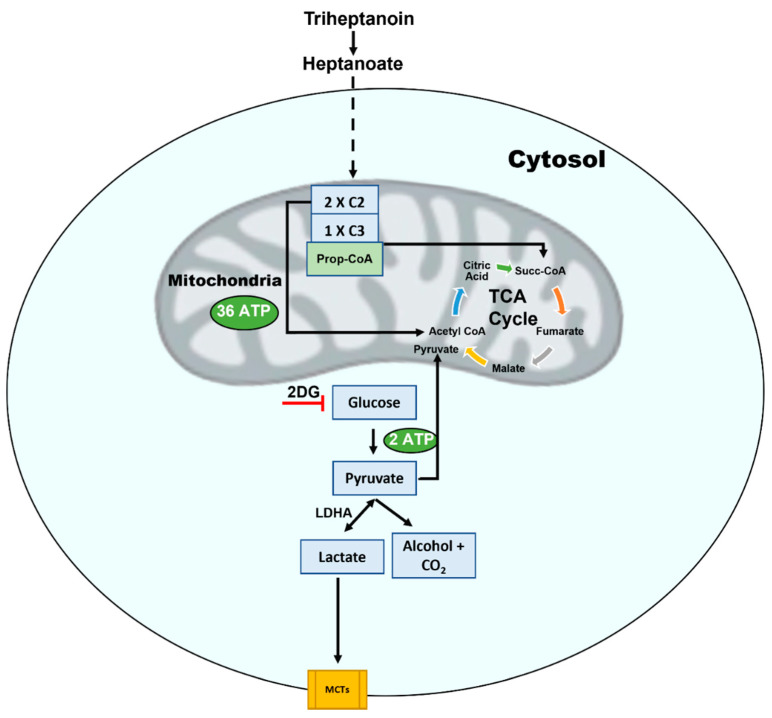
Triheptanoin, triglyceride of heptanoate replenishes the tricarboxylic acid (TCA) cycle via the propionyl–CoA carboxylase pathway. Following carboxylation, propionyl–CoA converts to succinyl–CoA. Further, heptanoate that enters the cell is metabolized into acetyl–CoA, which acts as an anaplerotic fuel for the synthesis of citric acid. Lactate produced by glycolysis is transported across the cell membrane by co-transport with protons and is mediated by monocarboxylate transporters (MCTs). Thus, triheptanoin can significantly enhance mitochondrial ATP production. 2 deoxy glucose (2DG), an inhibitory analogue of glucose, inhibits the glycolytic pathway causing metabolic stress to cells. A-T cells are hypersensitive to metabolic stress, which highlights the crucial role of ATM in mitochondrial metabolism and glucose homeostasis.

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
