# Peer review of "Metabolic Stress and Mitochondrial Dysfunction in Ataxia-Telangiectasia"

_antioxidants, 2022, doi:10.3390/antiox11040653_

Round 1

Reviewer 1 Report

I want to congratulate the authors for this review describing the role of TMJ in protecting cells from both external and internal damage. I find it complete in every detail and in every area on a cellular level.

Author Response

This reviewer did not raise any issues to be addressed.

Reviewer 2 Report

The present review by Goutham Narayanan Subramanian and colleagues, “Metabolic stress and mitochondrial dysfunction in ataxia-telan-2 giectasia”, authors describe the protein kinase ataxia-telangiectasia mutated (ATM) and its role in different diseases, and mechanism how ATM plays important role in cell protection, against various stresses to mitochondrial DNA damages.

In general, the review is interesting and potentially helpful for clinical as well as basic science understanding.  I have the following  comments:

  1. ATM’s role in diabetes is also very well established, including some references would likely improve the readability and impact of the review (here is one suggested reference, can be included- Helena Donath et al. 2020- frontiers in Pediatrics-Diabetes in Patients with Ataxia Telangiectasia: A national Cohort Study).
  2. Please check the sentence on page number 7, lines 255- 257 “A defect in mitophagy and the increased mitochondrial content reported for A-T cells would be expected to create an imbalance in copy number and the persistence of damaged mitochondrial that would interfere with normal mitochondrial”,  the sentence reads incomplete, or is it “mitochondrion”, mitochondrial DNA or mitochondrial function?
  3. Please replace Figures 1-5, with high resolution and clarity images, because figures are blurred at some parts are not clear. Figure 6 is not very well described, after conversion to lactate there must be some transporter to end or proceed the whole cycle?

Author Response

Reviewer 2 

  1. ATM’s role in diabetes is also very well established, including some references would likely improve the readability and impact of the review (here is one suggested reference, can be included- Helena Donath et al. 2020- frontiers in Pediatrics-Diabetes in Patients with Ataxia Telangiectasia: A national Cohort Study).

         We have inserted several sentences on insulin resistance and diabetes.                 Section 6 page 8 , lines 348 – 353.

  1. Please check the sentence on page number 7, lines 255- 25 the sentence reads incomplete, or is it “mitochondrion”, mitochondrial DNA or mitochondrial function?

         We have corrected this to “mitochondrial function” in updated manuscript            in section.

  1. Please replace Figures 1-5, with high resolution and clarity images, because figures are blurred at some parts are not clear.

          Higher resolution figures are provided.

  1. Figure 6 is not very well described, after conversion to lactate there must be some transporter to end or proceed the whole cycle?

         The figure legend has been amended to cover this.

Reviewer 3 Report

In this manuscript, Subramanian et al review the role of ATM in protecting the cell against both external and endogenous damage.

The review is well written, and the figures are very illustrative.

My suggestion is the inclusion of a section about the role of ATM in inflammation and inflammasome activation.

Author Response

  1. My suggestion is the inclusion of a section about the role of ATM in inflammation and inflammasome activation.

           We have inserted a description of this under section 4. ATM and                            inflammation.